# Automatic Extraction of Adverse Drug Reactions from Summary of Product Characteristics

Zhengru Shen [1,*] and Marco Spruit [2,3]

[1] Department of Information and Computing Sciences, Utrecht University, 3584 CS Utrecht, The Netherlands
[2] Department of Public Health and Primary Care, Leiden University Medical Center,
2511 DP The Hague, The Netherlands; m.r.spruit@lumc.nl
[3] Leiden Institute of Advanced Computer Science, Leiden University, 2333 CA Leiden, The Netherlands
[*] Correspondence: z.shen@uu.nl; Tel.: +31-30-253-6454

**Abstract:** The summary of product characteristics from the European Medicines Agency is a reference document on medicines in the EU. It contains textual information for clinical experts on how to safely use medicines, including adverse drug reactions. Using natural language processing (NLP) techniques to automatically extract adverse drug reactions from such unstructured textual information helps clinical experts to effectively and efficiently use them in daily practices. Such techniques have been developed for Structured Product Labels from the Food and Drug Administration (FDA), but there is no research focusing on extracting from the Summary of Product Characteristics. In this work, we built a natural language processing pipeline that automatically scrapes the summary of product characteristics online and then extracts adverse drug reactions from them. Besides, we have made the method and its output publicly available so that it can be reused and further evaluated in clinical practices. In total, we extracted 32,797 common adverse drug reactions for 647 common medicines scraped from the Electronic Medicines Compendium. A manual review of 37 commonly used medicines has indicated a good performance, with a recall and precision of 0.99 and 0.934, respectively.

**Keywords:** information extraction; natural language processing; adverse drug reactions; summary of product characteristics





## 1. Introduction

Drug product labels are regulatory documents required as part of the marketing authorization of each medicine. They provide up-to-date and comprehensive information about the risks, benefits, and pharmacological properties of marketed medicines. As such, extracting the clinical knowledge stored in product labels and making it available in the form of computationally accessible knowledge bases would benefit several applications in the area of drug safety surveillance and assessment. For example, during post-marketing safety assessments, it is crucial to determine whether an investigated adverse drug reaction (ADR) is already labeled [1,2].

There are two main versions of drug product labels around the world: Structured Product Labels (SPL) introduced by the Food and Drug Administration (FDA) in the US, and the Summary of Product Characteristics (SmPC) supervised by the European Medicines Agency (EMA) in the EU [3]. Both product labels provide information in sections, and the content of each section is in free narrative texts. This study focuses on the undesirable side effect section of product labels which describes adverse drug reactions (ADRs).

To date, there are no structured machine-readable ADRs. Therefore, extracting ADRs from unstructured product labels becomes an interesting research topic. There are many studies focusing on transforming unstructured ADRs data into structured machine-readable data [4–8]. However, the majority of current studies focus on the US version of product labels. For example, systems such as SPLICER and SPL-X were developed to extract ADR terms from particular sections using various natural language processing

(NLP) techniques, including named entity recognition, rule-based parsing, and NegEx [4]. Fung et al. proposed to use open-source NLP tools to extract drug indication information from SPL [5]. Wu et al. managed to fetch ADR terms using Oracle Text search from 1164 single-ingredient medicines [6]. A variety of NLP techniques were designed to extract ADRs from SPL in the 2017 Text Analysis Conference track [2].

However, the FDA SPL is the main data source used in such studies. Until now, the SmPC, as the European equivalent to SPL, has barely been investigated and no method has been developed solely for extracting ADRs from the SmPC. This study contributes to the field as a starting point.

In this study, the main research question we are addressing is how to utilize NLP techniques to automatically extract ADRs from standardized European product labels, namely SmPC. To answer the question, we first develop an NLP pipeline to extract adverse drug reactions from SmPC. A knowledge base is created from the extracted terms. The main characteristics of the NLP pipeline and the knowledge base are summarized as follows: (a) open-source and reproducible; (b) customizable for related ad hoc tasks; (c) knowledge base applicable for clinical studies.

The paper is structured as follows. Section 2 illustrates a number of methods and materials used in our study. Then the results are presented in Section 3. Discussions in Section 4 summarize the main findings of this research and some limitations. Section 5 concludes the study.

## 2. Materials and Methods

In this section, we describe the SmPC data source, elaborate on the various NLP techniques that are applied to extract ADRs from SmPC, and briefly explain the evaluation setup for validating our automatic ADR extraction approach. A conceptual overview of the ADR extraction pipeline is illustrated in Figure 1. The first step is to scrape the side effects data from the Electronic Medicines Compendium (EMC). Then the ADRs are extracted accordingly. Finally, the performance of our NLP method is evaluated by both NLP and clinical experts.

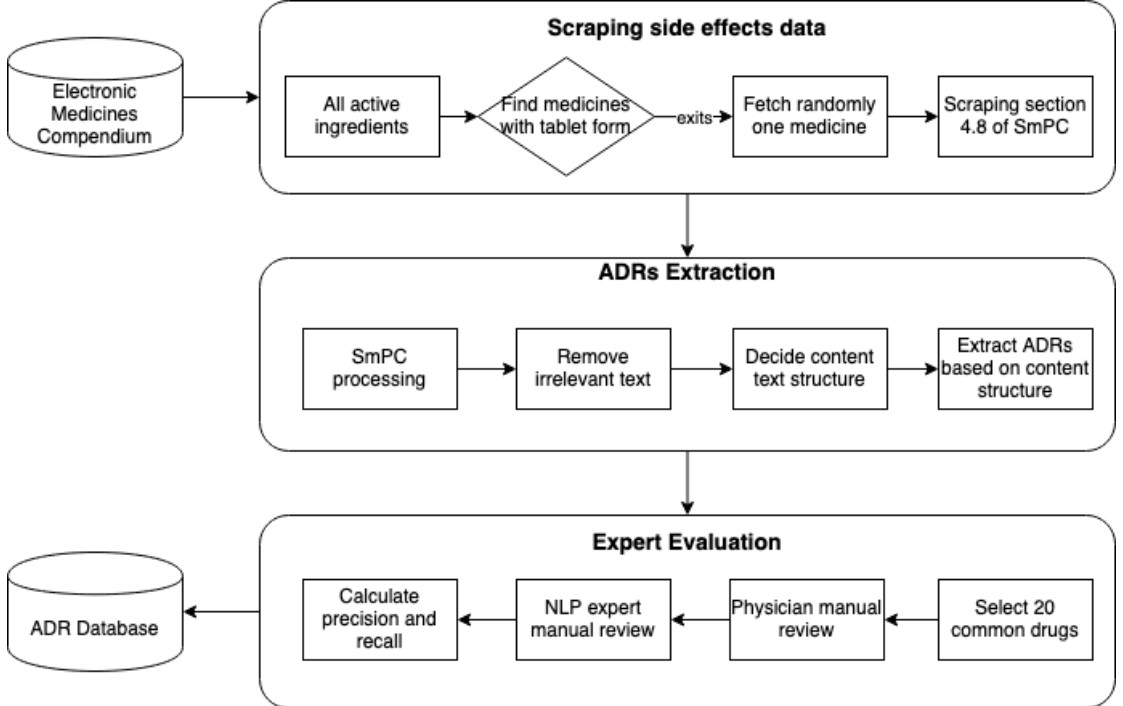

**Figure 1.** A conceptual overview of the automatic Adverse Drug Reactions (ADRs) extraction pipeline.

## 2.1. Dataset

All SmPCs were scraped from the EMC which provides the most recent and openly accessible regulated and approved information about medicines licensed in the UK [9]. The EMC was selected as our data source for two main reasons: (1) all information is in English; (2) it has more than 14,000 documents, all of which have been checked and approved by the European government agencies. Figure 2 shows an example of the side effects section of SmPC.

---

**4.8 Undesirable effects**

Summary of the safety profile

Headache, abdominal pain, diarrhoea and nausea are among those adverse reactions that have been most commonly reported in clinical trials (and also from post-marketing use). In addition, the safety profile is similar for different formulations, treatment indications, age groups and patient populations. No dose-related adverse reactions have been identified.

*Tabulated list of adverse reactions*

The following adverse drug reactions have been identified or suspected in the clinical trials programme for esomeprazole and post-marketing. None was found to be dose-related. The reactions are classified according to frequency (very common > 1/10; common ≥1/100 to <1/10; uncommon ≥1/1000 to <1/100; rare ≥1/10000 to <1/1000; very rare <1/10000); not known (cannot be estimated from the available data).

**Blood and lymphatic system disorders**

Rare: Leukopenia, thrombocytopenia

Very rare: Agranulocytosis, pancytopenia

**Immune system disorders**

Rare: Hypersensitivity reactions e.g. fever, angioedema and anaphylactic reaction/shock

**Metabolism and nutrition disorders**

Uncommon: Peripheral oedema

Rare: Hyponatraemia

Not known: Hypomagnesaemia (see section 4.4); severe hypomagnesaemia can correlate with hypocalcaemia. Hypomagnesaemia may also be associated with hypokalaemia

**Psychiatric disorders**

Uncommon: Insomnia

Rare: Agitation, confusion, depression

Very rare: Aggression, hallucinations

---

**Figure 2.** An example of a Summary of Product Characteristics (SmPC) excerpt.

## 2.2. Scraping the Side Effects Section of SmPC

Since this study focuses on commonly used medications, we only scrape the SmPCs of a limited number of medicines in the EMC. As shown in Figure 1, the process of scraping side effects data starts with identifying active substances. Then, for each active substance, only one medication in tablet form is included in the final list which contains 647 medicines, because the tablet form is the most common medication route. When there is no medicine in tablet form for a given active substance, none is selected. At last, section 4.8 Undesirable effects of the SmPC is obtained for all 647 medications in the final list. The 4.8 section consists of ADR information in both plain text and structured tables, we scraped the available information in HTML format so that the data structure is kept. All data is stored as a JSON file, which is used in the later steps.

## 2.3. ADR Extraction

### 2.3.1. SmPC Processing

In the second phase of our NLP pipeline is ADR extraction. It starts with processing the above-mentioned ADR relevant HTML excerpts. As we described before, the section 4.8 in the SmPC contains ADR information in diverse formats: plain text, structured text,

and tables. Tables used in describing ADR information also come with different structural styles. Therefore, we classify the 4.8 section into separate categories in terms of its content structure. Figure 3 elaborates on the process of categorization. Each category has its own ADR extraction technique.

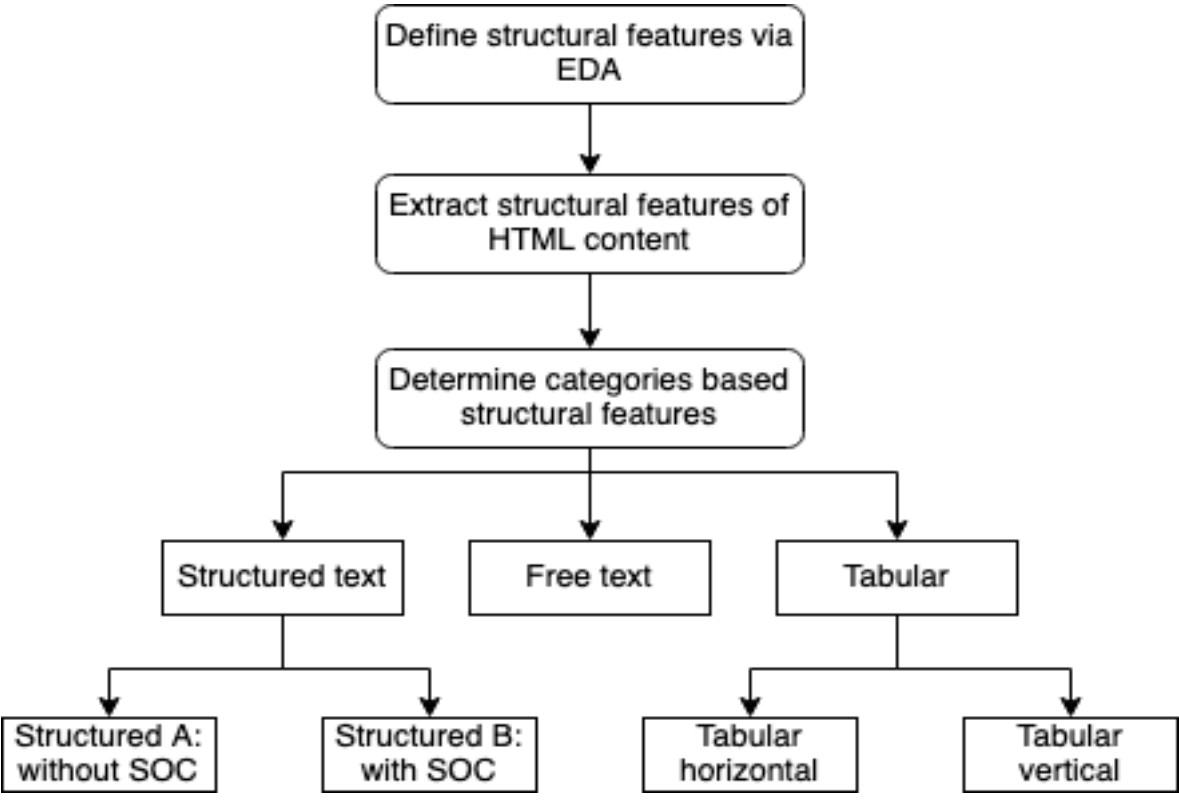

**Figure 3.** A decision tree for classifying the ADR content during SmPC processing.

First, we identify a list of structural features in the HTML excerpts, for instance, the counts of '<table>', the counts and positions of some MedDRA terms [10], such as the frequency and SOC (System Organ Classes) [11]. Table 1 shows some specific examples of such words. Secondly, the features are engineered from the scraped HTML files. Based on the features, we obtain three main structural categories: structured text, free text, and tabular.

- Free-text: ADRs are explained in free-text, as shown in Figure 4a. ADRs are hidden in sentences, which thus requires named entity recognition (NER) to extract them.
- Structured text: ADRs are described in a structured text, as shown in Figure 4b. It depicts a list of ADRs and their frequencies in an organized format.
- Tabular: ADRs are presented in a very structured way, namely a table. Figure 4c shows an example of such tables. These tables are further split into two groups: tabular horizontal and tabular vertical.

**Table 1.** Examples of ADR-specific terminologies.

| Term Type | Description | Examples |
|---|---|---|
| Frequency | Terms that describe how often an ADR happens | Very common<br>Common<br>Uncommon<br>Rare<br>Very rare<br>Not known |
| SOC | The highest level of groupings in MedDRA. There are 27 groups which are defined by etiology [9]. | Immune system disorders<br>Metabolism and nutrition disorders<br>Nervous system disorders<br>Eye disorders<br>Cardiac disorders<br>Gastrointestinal disorders<br>Skin and subcutaneous tissue disorders<br>General disorders and administration site conditions<br>Investigations |

**Figure 4.** Example fragments of the three main structural categories: (**a**) free text, (**b**) structured text, and (**c**) tabular text.

### 2.3.2. ADR Extraction

In the previous section, we identify three main structural categories that require different ADR extraction techniques, respectively. An overview of the extraction pipeline for each category is illustrated in Figure 5.

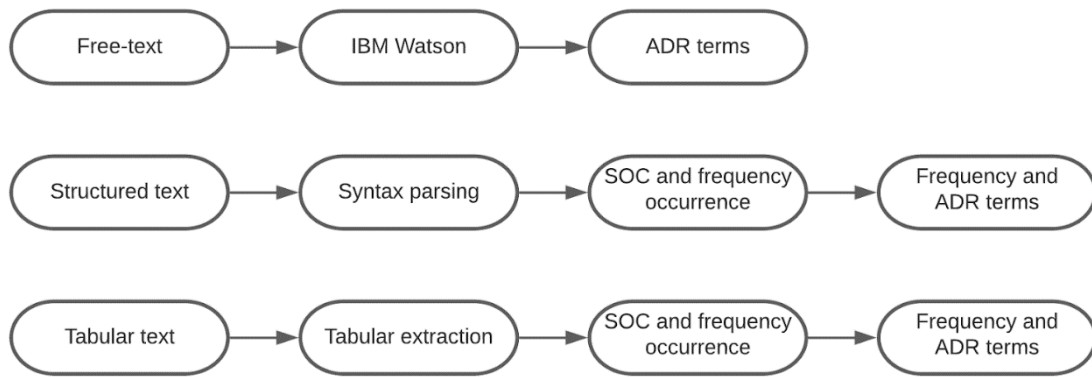

**Figure 5.** An Overview of the ADR Extraction Pipeline.

To uncover the hidden ADRs in the free-text, entity extraction techniques are applied. Specifically, we simplify the extraction process by using the IBM Watson Natural Language Understanding API that offers sophisticated NLP techniques in extracting meta-data from content such as concepts, entities, keywords, and others. Benchmarking studies have shown that IBM Watson Natural Language Understanding API is a simple and useful NLP tool in solving various clinical NLP problems [12,13]. A demonstration of how the API works is available [14]. Since the number of ADR terms is relatively limited, we can use the API for free. It offers a lite account which can process 30, 000 items per month free of charge [15]. Frequencies of the extracted ADRs are assigned as unknown due to their absence in the free texts.

For the structured text, we extract ADR terms and their corresponding frequencies with syntax parsing. The syntactic structure of a structured HTML text is depicted by the occurrences and positions of SOC (System Organ Class) and frequency terms. With the identified positions of SOC and frequency terms, we can obtain the HTML elements that contain only ADR terms. In most cases, ADRs terms in an HTML element are separated by a comma. Thus, by splitting the string by comma, we extract several ADRs for a given SOC and frequency. An example of such ADR extraction is illustrated in Figure 6. The left is the structured text in HTML and the right side shows the extracted ADRs in JSON.

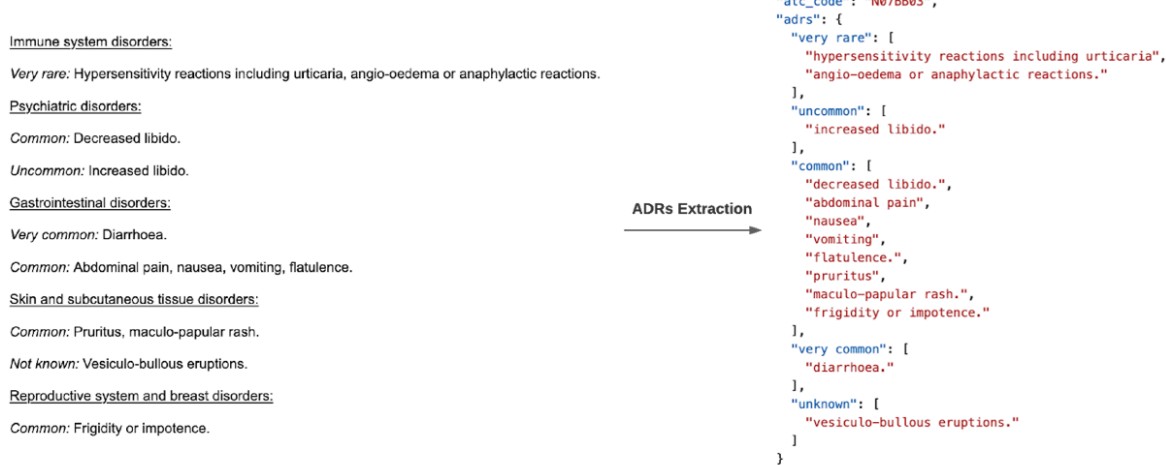

**Figure 6.** An example of ADR extraction for the structured text.

The tabular text refers to HTML tables that describe ADRs of a given medicine in a two-dimensional grid format. The first step of information extraction for HTML tables is to detect the structure. As mentioned above, we identify two types of structures: vertical and horizontal. Similar to structured text, ADR extraction starts with getting the occurrences and positions of SOC and frequency terms. Then, with the position indexes of SOC and

frequency terms, HTML table cells that contain ADR terms are fetched and processed. ADR terms in an HTML table cell are usually separated by a comma. An example is given in Figure 7.

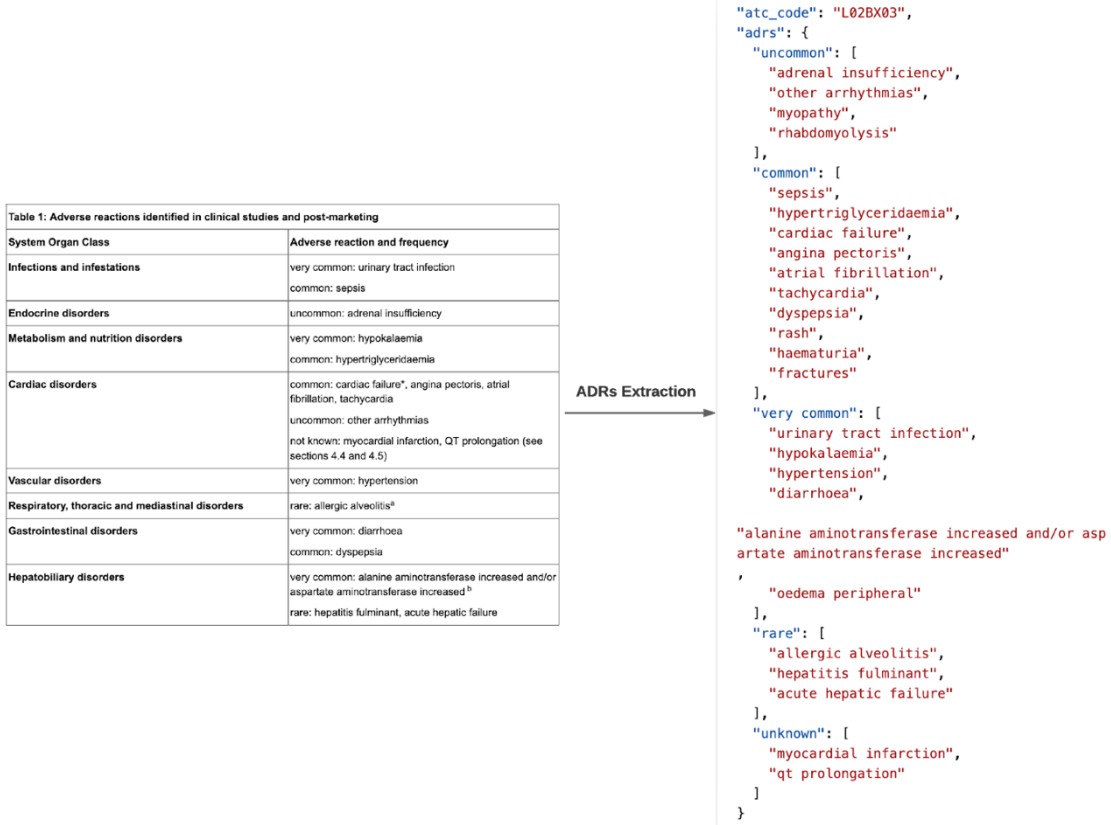

**Figure 7.** An example of ADR extraction for the tabular text.

### 2.4. Evaluation

An expert manual review is performed in the evaluation of ADRs extracted from the SmPCs in this study. To alleviate the burden of manual reviewing for our clinical experts, we only sample a small subset of common medicines. Moreover, an NLP expert is also involved in the manual review. The NLP expert specializes in processing the clinical text and has a sufficient clinical knowledge to properly review the extracted ADRS. In total, 37 medications were selected by the clinical expert for manual review. The clinical expert provided a list of top 50 commonly used medicines in Dutch primary care, and 37 of 50 were available in our scrapped drug list. At the beginning of the evaluation, both clinical and NLP experts are requested to review five identical medicines to align the manual reviews of different experts. Following that, due to time constraints, the clinical expert is asked to review five more medicines, whereas the NLP expert continues to review 32 more medicines.

For each medicine, all extracted ADRs are examined and labeled as 'correct', 'near', or 'incorrect'. Furthermore, the number of missing ADRs is counted by comparing the extracted ADRs with the original SmPC. Subsequently, we obtain the numbers of true positives (TP), false positives, and false negatives (FN). The ADR terms with only 'correct' label are considered as TP, both 'near' and 'incorrect' are FP, and the missing ADRs are FN. The standard metrics of recall, precision, and F-score are calculated to measure the performance of the proposed automated approach.

## 3. Results

This section presents the results of our proposed ADR extraction method and the manual evaluation results.

### 3.1. Overview

This study scrapes the SmPC documents of a total number of 647 marketed medicines, from which 32,797 ADR terms and their frequencies are extracted. Among all extracted ADR terms, 8069 are unique. The average number of ADR terms per medicine is 51. Table 2 offers a statistical overview of the extraction results from our selected medicines.

**Table 2.** Overview of the ADR extraction results.

| Characteristics | Statistics |
|---|---|
| # of selected marketed medicines | 647 |
| # of medicines with structured text | 141 |
| # of medicines with tabular text | 419 |
| # of medicines with free text | 87 |
| Average ADRs per medicine | 51 |
| 25% percentile ADRs per medicine | 21 |
| 50% percentile ADRs per medicine | 38 |
| 75% percentile ADRs per medicine | 67 |
| Top medicines in terms of extracted ADRs | 1. Topamax 100 mg Tablets (269)<br>2. Revolade 25 mg film-coated tablets (255)<br>3. Capecitabine Accord 150 mg film coated tablets (240)<br>4. Glivec 100 mg film-coated tablets (232)<br>5. Risperdal 0.5 mg Film-Coated Tablets (218)<br>6. Xadago 50 mg film-coated tablets (215)<br>7. Invega 12 mg prolonged-release tablets (207)<br>8. LUSTRAL 100 mg film coated tablets (207)<br>9. Isentress 100 mg chewable tablets (191) |

### 3.2. The Manual Evaluation Results

Clinical experts chose 37 commonly prescribed medicines for manual reviews. Appendix A provides a complete list of the 37 reviewed medicines. Among these chosen medicines, there are 28 tabular texts, 7 structured texts, and 2 free texts. As explained before, two experts participated in the manual evaluation. Table 3 shows the results of the manual review, including some important performance scores, such as recall and precision. For each medicine, the number of true positives, false positives, and false negatives are counted. An extracted term is considered as true positive only when it is reviewed as a correct ADR compared with the original text. False positives refer to the extracted terms that are not ADRs. The missing ADRs are labeled as false negatives. The overall recall and precision of all reviewed medicines are 0.99 and 0.932, respectively, which shows the effectiveness of our ADR extraction approach and the reliability of its outputs.

**Table 3.** Final results of the manual expert reviews.

| | Reviewer 1 (NLP Expert) | Reviewer 2 (Clinical Expert) | Totals |
|---|---|---|---|
| # of reviewed medicines | 32 | 5 | 37 |
| # of extracted ADRs terms | 1700 | 118 | 1824 |
| # of correct ADRs (TP) | 1590 | 110 | 1703 |
| # of incorrect ADRs (FP) | 116 | 8 | 124 |
| # of missing ADRs (FN) | 7 | 11 | 18 |
| Recall | 0.996 | 0.909 | 0.99 |
| Precision | 0.932 | 0.932 | 0.932 |

*3.3. Error Analysis*

Three main types of errors are identified among the 116 incorrect extracted ADRs of Reviewer 1 (false positives). Table 4 summarizes the error analysis of incorrectly extracted ADRs. Our approach encountered issues with splitting multiple ADR terms that are joined by a colon or semicolon. For instance, two ADR terms (agitation and aggression) could be extracted from 'Agitation, aggression'. However, if multiple ADR terms are joined by 'or', 'and', or semicolon, such as the examples shown in Table 4, our extraction method fails to properly address this. The second type of false-positive results is related to the data cleaning of our approach. The extracted ADRs contain noises such as unrelated words or special characters. As shown in the examples, to get the correct ADRs, we need to clean up noises like '\u2020', and 'in combination with insulin or sulphonylurea'. Lastly, a small number of extracted terms are not ADRs at all. For example, 'skin and subcutaneous tissue disorders' is an SoC term that should be excluded from our final ADR lists. Since other incorrect extractions do not belong to any specific group, it is difficult to exclude them. However, the number of such errors is so small that its impact is limited.

**Table 4.** Error analysis for false positives.

| Type of Error | Examples | | Counts |
|---|---|---|---|
| | **Extracted Terms** | **Correct ADRs** | |
| Unsplit multiple ADTs | "angio-oedema or anaphylactic reactions" | angio-oedema, anaphylactic reactions | 81 |
| ADR with noise | - 'retinal detachment\u2020'<br>- 'hypoglycaemia in combination with insulin or sulphonylurea' | retinal detachment, hypoglycaemia | 22 |
| Non-ADR | - 'all causality frequency',<br>- 'general disorders:'<br>- 'skin and subcutaneous tissue disorders' | Not ADR | 13 |

Since there are only seven missing ADRs out of 1706 extracted ADRs, false-negative (missing ADR) is not a common error. However, we found that such an error is more likely to happen to medicines using free text to describe their ADRs. Since only 87 out of the selected 647 medicines summarize their ADRs in free-text, and the number of unstructured reported medicines has been steadily decreasing with recent SmPC updates,

the false-negative error has a relatively limited and decreasing impact on the performance of our approach.

## 4. Discussion

Many studies investigated the extraction of ADRs from SPLs from the FDA in the USA, and some effective methods have been developed. However, as an equivalent in Europe, the SmPC from the European Medicines Agency receives very little attention in the field of NLP. In this study, we fill the research gap with the development of an automated ADRs extraction method for the SmPC.

The manual experts review results demonstrate that our proposed method is effective and has the potential of being used to solve ADR related clinical problems. Specifically, our method achieves an overall recall of 0.990 and a precision of 0.932. Such high performance has never been reported in previous studies focusing on extracting ADRs from SPLs [6,16].

As discussed in the error analysis, there are a few ways to further improve the performance of our method. For example, noise in the extracted terms can be cleaned with regular expressions. When it comes to the unsplit multiple ADRs, we can extend the split function with more options so that it allows strings to be split by 'or', 'and', or semicolon. Furthermore, encoding extracted ADR terms into MedDRA Preferred Terms could reduce the non-ADR errors, thus further improving the performance.

Our study has some limitations. First, there are no standardized ADR annotations based on the SmPC available to benchmark ADR extraction methods. It is difficult to reproduce the performance of this study. Manual reviews from different experts might present different performance scores.

Another limitation lies in the manual review process of this study. Involving clinical experts in the manual review process is always challenging given their busy schedule. The coronavirus has made the situation even worse. Therefore, the manual process in this study only included one clinical expert. To compensate for this, we added one clinical NLP expert as another reviewer. The NLP expert is familiar with this topic and was trained for the task by the clinical expert. To address this, we plan to expand the manual review to a larger size of samples and a bigger group of clinical experts in a post-COVID world.

The proposed ADR extraction method is developed based on data from the EMC in UK. However, due to Brexit, product labels in UK might changes. Then the method will not perform the same as shown in this study. Further development of this method should focus on using the SmPCs from the EMA.

## 5. Conclusions

The contributions of this study are two-fold. First, it contributes to the field of clinical NLP by introducing an open-source and reproducible method that extracts ADR terms from the SmPC. The high-performance scores (recall: 0.99 and precision: 0.934) indicate our approach is very effective, which leads us to believe that the method could be useful in the processing and coding of ADRs in the SmPC. The second contribution lies in the clinical field. The results of our extraction method are structured data that describes marketed medicines and their recorded ADRs and frequencies. Such a knowledge base could be applied to address practical clinical problems, ranging from ADR assessment in clinical trials to assisting the detection of ADRs in patients.

**Supplementary Materials:** The following materials are available online at https://github.com/ianshan0915/ade-extraction/paper, Appendix A: Appendix A-manual review results.xlsx. Source code of the project is available on https://github.com/ianshan0915/ade-extraction.

**Author Contributions:** Z.S. conducted the research, including the research design, data collection, and data analysis. M.S. supervised the research and reviewed the paper. All authors have read and agreed to the published version of the manuscript.

**Funding:** This work is part of the project "OPERAM: OPtimising thERapy to prevent Avoidable hospital admissions in the Multimorbid elderly", supported by the European Commission (EC)

HORIZON 2020, proposal 634238, and by the Swiss State Secretariat for Education, Research and Innovation (SERI) under contract number 15.0137. The opinions expressed and arguments employed herein are those of the authors and do not necessarily reflect the official views of the EC and the Swiss government.

**Institutional Review Board Statement:** Not applicable.

**Informed Consent Statement:** Not applicable.

**Data Availability Statement:** Data available in a publicly accessible repository. The data presented in this study are openly available in [ade-extraction] at [https://github.com/ianshan0915/ade-extraction].

**Acknowledgments:** We thank Bastiaan Sallevelt for helping us review the extracted ADRs.

**Conflicts of Interest:** The authors declare no conflict of interest.

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
