# Peer review of "Automatic Extraction of Adverse Drug Reactions from Summary of Product Characteristics"

_applsci, doi:10.3390/app11062663_

Round 1
Reviewer 1 Report
The paper is interesting and well written. The topic of ADR extraction from not well structured information could be useful and is clearly interesting. The validation methodology based mainly on a small sample of the results and performed mainly by a NLP specialist is quite poor, this should be much better justified. The availability of the code and the dataset is an important contribution.
Reviewer 2 Report
Good work! I just have two minor remarks:
- Line 130 "... API is a simple and useful NLP tool in solving various [11-13]." End part of this phrase is missing?
- Line 131. You write that IBM Watson is available for free due to the small number of ADR terms. Could you elaborate a little bit on this? What are those prerequisities to use IBM Watson? Any direct link to look into?
Reviewer 3 Report
The authors aim to develop an automatic natural language processing (NLP) technique to extract adverse drug reactions (ADRs) from drugs’ Summary of Product Characteristics (SmPC) since similar tools already exist for FDA Structured Product Labels but not for EMA SmPCs. The method developed by the authors showed a good accuracy on the 37 drugs’ SmPC tested. The authors developed an interesting tool and put their source code and data into open access. We have some minor comments regarding this manuscript that must be addressed, nonetheless.
Major points
- Data from Electronic Medecines Compendium have been used to identify ADRs from drugs’ SmPCs. UK is no longer part of the EMA though, since Brexit and their product labels have the flexibility to change in the near future. It would be wise to also develop a tool to extract ADR directly from EMA’s SmPC (https://www.ema.europa.eu/en/medicines) which are also in English.
- Only NLP experts, who somehow lack of medical formation, reviewed the majority of SmPCs. Thus, they may not identify similar ADRs and/or distinguish from overlapping ADRs. What were the instructions to code for “correct” (exact term?), “near” (similar term but medically different?) and “incorrect” (very different term?)?
Minor points
- The word “pharmacogenomics” is used several times in the “Introduction” part to refer to product label. That word seems a bit “off-topic” and might be misunderstood. The manuscript would gain from some clarifications on that point. Alternatively, the authors could use another word.
- The abstract mentions that the 37 SmPCs were randomly selected. However, in the “Results” part, it is stated that the SmPCs “were chosen by clinical experts” with no information given on the selection method in the “Methods” part. The authors must clarify the method used to select the 37 SmPCs.
- The authors choose SmPCs of tablet forms only, because it is the most common pharmaceutical form. However, other forms (i.e. parenteral) are associated with ADRs as well. By excluding them, the accuracy of the method is blunted for those pharmaceutical forms.
- Table 2 could be improved by indicating also the percentage of extracted ADRs per medicine.
- Information is lacking about the type of SmPC (structured text, tabular text, free text) for the 37 reviewed SmPCs.
- It is questionable to use another dictionary than the widely available and accepted MedRA (https://www.meddra.org/), which is the standardized ADR annotation used in pharmacovigilance (both by EMA, FDA and WHO). Why did the authors avoid MedRA? Do they have a good reason?
Round 2
Reviewer 3 Report
Point 6.
I apologize for the lack of clarity in my questioning. The authors gave only the number of ADRs yielded by their method without mentioning the total number of ADRs present in each SmPC, i.e. for Topamax if the total number of ADRs mentioned in the SmPC is 270, the resulting accuracy would be of 99%. Such accuracy would lower to 90% if 300 was the real number or to 53% if 500 was the absolute number. The authors must mention such percentages observed in their paper.
The paper is now greatly improved. One minor correction to add is the percentage of detected ADRs for the 9 drugs mentioned in table 2.
Author Response
First of all, thanks for the clarification and all the suggestions that you provide.
Here is my response to point 6.
We do not include the percentage of detected ADRs for the following reasons.
- The percentage of detected ADRs does not equal accuracy because the extracted ADRs terms from our method could be wrong. In that case, they are false positives. To know how many are false positives, we need to manually review each listed drug in Table 2. However, none of the drugs is included in our manual review.
- Again, without manual review, it's difficult to know the total number of ADRs in each SmPC. Therefore, the percentage of detected ADRs could not be calculated.
- The absolute numbers of extracted ADRs of the listed 9 drugs are intended to show the statistics overview of our results, not to demonstrate the performance of our method.
- The accuracy per drug you refer to can be found in the supplementary materials, which are downloadable via this link.
Given the above reason, we think it's better not to add the percentages. We hope the answer is clear to you.